# KITE: Kernelized and Information Theoretic Exemplars for In-Context Learning

## Abstract

In-context learning (ICL) has emerged as a powerful paradigm for adapting large language models (LLMs) to new and data-scarce tasks using only a few carefully selected task-specific examples presented in the prompt. However, given the limited context size of LLMs, a fundamental question arises: Which examples should be selected to maximize performance on a given user query? While nearest-neighbor-based methods like KATE have been widely adopted for this purpose, they suffer from well-known drawbacks in high-dimensional embedding spaces, including poor generalization and a lack of diversity. In this work, we study this problem of example selection in ICL from a principled, information theory-driven perspective. We first model an LLM as a linear function over input embeddings and frame the example selection task as a query-specific optimization problem: selecting a subset of exemplars from a larger example bank that minimizes the prediction error on a specific query. This formulation departs from traditional generalization-focused learning theoretic approaches by targeting accurate prediction for a specific query instance. We derive a principled surrogate objective that is approximately sub-modular, enabling the use of a greedy algorithm with an approximation guarantee. We further enhance our method by (i) incorporating the kernel trick to operate in high-dimensional feature spaces without explicit mappings, and (ii) introducing an optimal design-based regularizer to encourage diversity in the selected examples. Empirically, we demonstrate significant improvements over standard retrieval methods across a suite of classification tasks, highlighting the benefits of structure-aware, diverse example selection for ICL in real-world, label-scarce scenarios. We conduct extensive experiments over multiple classification datasets using several LLM models and demonstrate significantly improved performance achieved by our exemplar retrieval algorithm.

## 1 Introduction

With the advent of highly capable large language models (LLMs) (Adiwardana et al., 2020; Wang et al., 2019; Zhang et al., 2021; Wang et al., 2022), in-context learning (ICL) (Rubin et al., 2022; Liu et al., 2022; Wu et al., 2022) via prompt optimization has emerged as a powerful technique for generating responses to complex user queries in data-scarce settings. In this popular and practical paradigm, we assume access to a small bank of high-quality task-specific examples. For a given user query, a few unique and relevant demonstrations are selected to form additional context for the language model. Since LLMs are pre-trained on large corpora, even a small number of carefully chosen exemplars can often suffice to guide the model toward producing accurate and task-consistent responses (Luo et al., 2024). The key is to select examples that are both representative and query-relevant, enabling the model to implicitly infer the task. Compared to fine-tuning (see Wang et al. (2025) and references therein), ICL offers a lightweight and efficient alternative, especially when labeled data is limited.

Given the limited context window of large language models, a natural and important question arises for ICL: *"Given a specific user query, how can we optimally select and order a subset of task-specific examples from an associated example bank to include in the prompt to maximize performance?"* While several existing methods address this question empirically (Dong et al., 2022; Luo et al., 2024), our work takes a principled, information-theoretic approach to tackle this problem.

Motivated by data-scarce settings and the need for task generalization, we focus on common and practical scenarios where the exemplar retriever is frozen and non-trainable. Among unsupervised retrieval methods, the most widely used is KATE (Liu et al., 2021), which pioneered a $k$-nearest

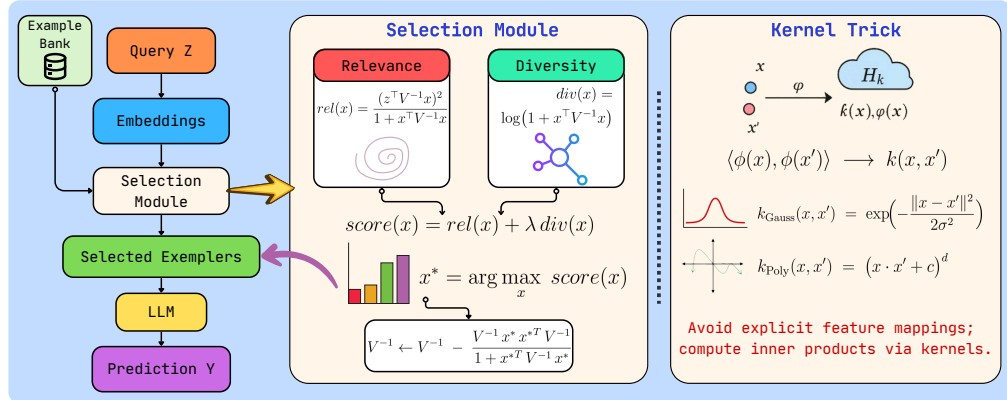

Figure 1: This figure is divided into two halves, illustrating the KITE's selection pipeline on the left and the kernelization trick on the right. **Left:** The selection module (Alg. 1) maintains an inverse design matrix $V^{-1}$ and, for each candidate $x$, computes relevance (Eq. (8)) and diversity (Eq. (10)), combines them into a total score (Eq. (11)), selects the highest-scoring example, and updates $V^{-1}$ via Sherman–Morrison (Eq. (12)). The chosen exemplars are then fed into the LLM to produce the final prediction $y$. **Right:** The kernel trick panel shows how every inner product in feature space is replaced by a kernel evaluation, $\langle \phi(x), \phi(x') \rangle \to k(x, x')$ (Lemma 1, Eq. (13)), enabling use of Gaussian RBF and polynomial kernels to work implicitly in the reproducing kernel hilbert space (RKHS) without ever computing high-dimensional feature vectors.

neighbor (kNN)-based strategy for selecting in-context examples. KATE identifies the $k$ examples most similar to the user query in a pre-trained embedding space (e.g., BERT), drawing on the intuition that nearby points in this space are likely to be the most informative.

However, as with classical kNN, this approach suffers from the curse of dimensionality—a well-documented issue in high-dimensional spaces where distances become less informative (Köppen, 2000). In traditional machine learning, this problem is often addressed by assuming a structured hypothesis class (e.g., linear models), which effectively reduces the model complexity and improves generalization guarantees with fewer samples. This motivates our central theoretical question in the ICL context: *"Can we design an example selection algorithm—i.e., a retriever—that, given a user query, operates under a structured modeling assumption to improve selection quality?"* Such an information-theoretic modeling-driven approach can not only mitigate the limitations of high-dimensional retrieval but also implicitly encourage diversity among selected examples - another desirable property for generalization in ICL.

From a theoretical standpoint, we model the LLM as a function that behaves linearly in its input, and the goal of ICL in the context of LLMs translates to algorithmically selecting a small number of datapoints to train the linear model (conditioned on an input test query).

Suppose we are given a high-quality example bank of $n$ labeled datapoints. For a specific $d$-dimensional user query $\mathbf{z}$, we consider the following core problem: *"Given a test point $\mathbf{z}$ and an example bank $(\mathbf{x}_i, y_i)_{i=1}^n \in \mathbb{R}^d \times \mathbb{R}$, which $k$ examples should be selected to train a linear model such that the prediction error on $z \in \mathbb{R}^d$ is minimized?"* Note that a selection algorithm for the aforementioned problem can be directly used for selecting ICL examples for LLMs. [1]

Theoretically speaking, this departs from standard learning-theoretic goals, which typically aim for generalization over a distribution of test points. In contrast, we focus on query-specific generalization, i.e., training a model that performs well on a single (and potentially arbitrary) test query using a carefully selected subset of training examples. For the linear model, we derive a surrogate loss objective that quantifies this query-specific prediction error and show that this objective is approximately submodular. This key structural insight allows us to apply a greedy selection algorithm with a provable $1 - e^{-\gamma}$ approximation guarantee relative to the optimal subset - here $\gamma \in (0, 1)$ is the approximate sub-modularity ratio of the objective given the example bank.

Building on this formulation, we design a computationally efficient and fully unsupervised example selection algorithm. We evaluate it across multiple tasks and find that it consistently outperforms strong

---

[1]In fact, if we were using kNN to predict the response for $\mathbf{z}$, then the selected datapoints would have been the $k-$nearest datapoints to $\mathbf{z}$. This is also the intuition for choosing examples in KATE (Liu et al., 2021).

baselines such as kNN-based top-k retrieval (Liu et al., 2021), DPP-based retrieval (Ye et al., 2023a), and BM25 (TF–IDF) keyword retrieval, demonstrating the effectiveness of our theory-guided retrieval in in-context learning. For example, our algorithm surpasses the strongest baseline, DPP (Ye et al., 2023a) by 2.44% on HellaSwag benchmark (Zellers et al., 2019) with Qwen-2.5-1.5B model (Hui et al., 2024). In this study, our focus is primarily on classification tasks, where the output is a single label, aligning with the assumptions of our theoretical framework. Note that classification tasks encompass a large fraction of use-cases where LLMs are used in a widespread fashion; for instance, LLM-as-a-judge (Zheng et al., 2023), toxicity detection (Gehman et al., 2020), and intent detection (Larson et al., 2019).

In addition to our base algorithm, we incorporate two complementary techniques to further enhance example selection. First, in the theoretical setting, we extend our greedy algorithm under linear models to the general nonlinear models using the well-known kernel trick. This allows our method to operate in high (possibly infinite) dimensional feature spaces without explicitly computing the feature mappings. By replacing inner products in the objective with kernel evaluations, we enable richer representations that capture nonlinear relationships between datapoints. Empirically, we observe that using a well-calibrated kernel can lead to noticeable improvements in LLM performance, suggesting that the choice of feature space plays a critical role in guiding in-context learning. Second, we introduce a mechanism for enforcing diversity in the selected examples. Inspired by maximum information gain theory—a well-established framework in experimental design, bandits, and Bayesian optimization (Lattimore & Szepesvári, 2020) — we consider the scenario where the goal is to select a subset of examples such that the resulting trained model performs well across all queries in the example bank. This naturally leads to the classical optimal experimental design problem, whose objective function is well-understood. We incorporate this design objective as a regularizer into our selection criterion to encourage diversity among selected examples. Intuitively, and as supported by our experiments, promoting diversity improves the generalizability of the model and boosts the quality of LLM responses.

## 2 RELATED WORKS

**In-Context Learning (ICL).** In-context learning, introduced by Brown et al. (2020b), is a powerful paradigm where large language models (LLMs) learn new tasks by conditioning on a few input-output examples provided in the prompt, without any parameter updates. The underlying mechanisms of ICL have been extensively studied; some works suggest that ICL enables the prediction task to become linearly separable (Saunshi et al., 2020) or that it allows the model to infer a shared latent concept from demonstrations (Xie et al., 2021). Further research indicates that ICL is a nuanced process, as models do not always rely strictly on the provided input-output mappings (Min et al., 2022). The ICL capabilities of LLMs can be enhanced through dedicated self-supervised or supervised training procedures (Chen et al., 2022; Min et al., 2021; Wei et al., 2023a). A broad range of studies has also investigated the factors that influence ICL performance, such as demonstration calibration and corpora effects (Zhao et al., 2021; Shin et al., 2022; Wei et al., 2022; Yoo et al., 2022; Wei et al., 2023b), as well as the working mechanisms themselves, including how models perform implicit gradient descent (Olsson et al., 2022; Li et al., 2023b; Pan, 2023; Dai et al., 2022).

**Example retrieval for few-shot learning.** The effectiveness of ICL is highly sensitive to the choice of in-context examples, a variance quantified in early works (Brown et al., 2020b; Min et al., 2022), which has motivated a significant line of research on demonstration selection. Initial approaches focused on unsupervised heuristics, such as retrieving semantically similar examples using embedding-based nearest neighbors (Liu et al., 2022) or lexical overlap, which showed sizeable gains over random sampling. Subsequent research has advanced beyond fixed heuristics to instead learn a retriever model. These methods often fine-tune a dense retriever to select effective demonstrations, using labels distilled from a language model (Rubin et al., 2022) or employing reinforcement and contrastive objectives to balance relevance and coverage (Li et al., 2023a; Luo et al., 2023; Wang et al., 2023; Ye et al., 2023b; Ghosal et al., 2025b;a). Orthogonal to retrieval, information-theoretic criteria such as mutual information (Sorensen et al., 2022) or LM perplexity (Gonen et al., 2023) serve as lightweight proxies for example quality. Influence-function analysis by Li & Qiu (2023) selects training points that exert the greatest effect on the language model's prediction. The current state-of-the-art involves subset-level methods that explicitly model interactions between examples, often using determinantal point processes (DPPs) to promote diversity and avoid redundancy (Yang et al., 2023; Ye et al., 2023a).

Based on an information-theoretic standpoint, in this paper, we propose a computationally efficient and fully unsupervised example selection framework.

## 3 THEORETICAL MODEL

Consider access to a dataset $\mathcal{X}$ containing $n$ known input-output pairs $(\mathbf{x}_1, y_1), (\mathbf{x}_2, y_2), \ldots, (\mathbf{x}_n, y_n) \in \mathbb{R}^d \times \mathbb{R}$, where features are $d$-dimensional vectors and the output are real scalars.

Further, consider an input test query $\mathbf{z} \in \mathbb{R}^d$. For any $k \ll d$, the ICL problem is defined as selecting the optimal subset of examples from $\mathcal{X}$ to provide as context (to the LLM) or train a model for predicting the response to the input test query $\mathbf{z}$. The goal is to ensure that the prediction error is minimized via an optimal choice of examples. Note that in ICL, the selected subset contains unique elements and cannot be a multi-set. This is because, in an LLM, duplicating the examples does not provide additional context.

To make the problem mathematically tractable, conditioned on the input test query $\mathbf{z} \in \mathbb{R}^d$, assume that the model response (in this instance, the LLM) is linear in its input with a distinct underlying parameter vector $\theta \in \mathbb{R}^d$. The selected subset of $k$ examples is used to train the model to generate a prediction for the test query $z \in \mathbb{R}^d$.[2] Specifically, conditioned on $\mathbf{z}$, there exists an unknown parameter vector $\boldsymbol{\theta} \in \mathbb{R}^d$ such that the response $y$ for any feature vector $\mathbf{x} \in \mathbb{R}^d$ is generated as

$$y = \langle \mathbf{x}, \boldsymbol{\theta} \rangle + \eta, \text{ where } \eta \sim \mathcal{N}(0, 1) . \tag{1}$$

In the above framework, the ICL problem reduces to the following question: *For a given test query $\mathbf{z}$, which subset of examples $\mathcal{S}$ of size at most $k$ from $\mathcal{X}$ should we select so that a least squares estimator fitted on $\mathcal{S}$ minimizes the expected error for the prediction on $\mathbf{z}$?* Mathematically, given a test query $\mathbf{z}$, we aim to solve the optimization problem:

$$\min_{\mathcal{S} \subseteq \mathcal{X} : |\mathcal{S}| \leq k} |\langle \mathbf{z}, \boldsymbol{\theta} - \widehat{\boldsymbol{\theta}}_{\mathcal{S}} \rangle| , \tag{2}$$

where $\widehat{\boldsymbol{\theta}}_{\mathcal{S}} = \left( \beta \mathbf{I}_d + \sum_{i \in \mathcal{S}} \mathbf{x}_i \mathbf{x}_i^\top \right)^{-1} \left( \sum_{i \in \mathcal{S}} \mathbf{x}_i y_i \right)$ is the $\beta$-regularized least squares estimator.

This discrete optimization problem is computationally challenging, with complexity that is exponential in $k$ and polynomial in $n$, stemming from the combinatorial nature of selecting subsets—specifically, the $\binom{n}{k}$ possible combinations. Note that this poses a severe computational challenge since this runtime will be required per user query. On the other hand, fast algorithms during inference without hurting the accuracy are of paramount importance. Our goal is to develop an efficient algorithm that can find high-quality solutions to the optimization problem in equation 2.

## 4 ALGORITHM DESIGN

We develop a principled approach to solve the in-context learning subset selection problem by leveraging submodular optimization theory (Das & Kempe, 2011). Our methodology consists of three main components: problem reformulation using concentration inequalities, theoretical analysis of submodularity, and a greedy algorithm with provable approximation guarantees.

**Problem Reformulation** To make the optimization problem in equation 2 tractable, we use concentration inequalities to bound the prediction error. Applying the Chernoff bound for sub-Gaussian random variables, we can bound, for any $\delta \in (0, 1)$, the prediction error as

$$|\langle \mathbf{z}, \boldsymbol{\theta} - \widehat{\boldsymbol{\theta}}_{\mathcal{S}} \rangle| \lessapprox \sqrt{\|\mathbf{z}\|_{\mathbf{V}_{\mathcal{S}}^{-1}}^2 \log(1/\delta)} , \tag{3}$$

with probability greater than $1 - \delta$, where $\mathbf{V}_{\mathcal{S}} = \beta \mathbf{I}_d + \sum_{i \in \mathcal{S}} \mathbf{x}_i \mathbf{x}_i^\top$ is the design covariance matrix, and a constant $\beta$ dependent factor is hidden under $\lessapprox$. This is a well-known result on the prediction error in linear models - for instance, see (Lattimore & Szepesvári, 2020, Chapter 20).

The above theoretical bound shows that the upper bound on prediction error is related to the chosen subset $\mathcal{S}$ of training datapoints via the term $\|\mathbf{z}\|_{\mathbf{V}_{\mathcal{S}}^{-1}}^2$. Hence, we need to minimize $\|\mathbf{z}\|_{\mathbf{V}_{\mathcal{S}}^{-1}}^2$ for a fixed

---

[2]We emphasize that the underlying parameter vector $\theta \in \mathbb{R}^d$ can vary conditioned on the test query $\mathbf{z}$.

$\mathbf{z} \in \mathbb{R}^d$, or, equivalently solve the following optimization problem

$$\max_{\mathcal{S} \subseteq \mathcal{X}:|\mathcal{S}|\leq k} f_{\mathbf{z}}(\mathcal{S}) \,, \text{ where } f_{\mathbf{z}}(\mathcal{S}) = -\mathbf{z}^\top \mathbf{V}_{\mathcal{S}}^{-1}\mathbf{z} \,. \tag{4}$$

The set function $f_{\mathbf{z}}$ is monotonically increasing, which follows from the fact that for any two sets $\mathcal{S} \subseteq \mathcal{L}$, the corresponding design matrices satisfy $V_{\mathcal{S}} \preceq V_{\mathcal{L}}$ in the Lowener order. Along with monotonicity, a desirable property for maximizing set functions is submodularity, defined as:

**Definition 1** (Submodular Function). *A set function $f : 2^{\mathcal{X}} \to \mathbb{R}$ is called* submodular *if for all $\mathcal{S} \subseteq \mathcal{L} \subseteq \mathcal{X}$ and $\mathbf{x} \in \mathcal{X} \setminus \mathcal{L}$, it satisfies the diminishing returns property:*

$$f(\mathcal{S} \cup \{\mathbf{x}\}) - f(\mathcal{S}) \geq f(\mathcal{L} \cup \{\mathbf{x}\}) - f(\mathcal{L}) \,.$$

For monotone and submodular functions, a greedy algorithm admits an $(1-1/e)$-factor approximation.

The set function $f_{\mathbf{z}}$ in equation 4 doesn't satisfy sub-modularity. However, it exhibits approximate submodularity and retains the near-optimality guarantee of greedy algorithms. The notion of approximate submodularity can be captured by the *submodularity ratio* Das & Kempe (2011), formally defined below.

**Definition 2** (Submodularity ratio). *For a ground set $\mathcal{X}$ and a parameter $k \in \mathbb{N}$, the submodularity ratio of a set function $f : 2^{\mathcal{X}} \to \mathbb{R}$ is defined as*

$$\gamma_k(f) = \min_{\substack{\mathcal{S} \subseteq \mathcal{X}, \\ \mathcal{L} \subseteq \mathcal{X}:|\mathcal{L}|\leq k, \\ \mathcal{L} \cap \mathcal{S} = \emptyset}} \frac{\sum_{\mathbf{x} \in \mathcal{L}} (f(\mathcal{S} \cup \{\mathbf{x}\}) - f(\mathcal{S}))}{f(\mathcal{S} \cup \mathcal{L}) - f(\mathcal{S})} \,.$$

The submodularity ratio captures how much more the value of the function $f$ can increase by adding any subset $\mathcal{L}$ of size at most $k$ to $\mathcal{S}$, compared to the combined benefits of adding its elements to $\mathcal{S}$. Hence, it quantifies the degree to which $f$ satisfies the diminishing returns property, and hence how well greedy algorithms are expected to perform in maximizing $f$ under cardinality constraints. If the set function $f$ is submodular, then $\gamma = 1$. When $0 < \gamma < 1$, the function is said to be approximately submodular. The next result lower bounds the submodularity ratio of $f_{\mathbf{z}}$.

**Lemma 1** (Submodularity ratio of $f_{\mathbf{z}}$). *For any $\mathbf{z} \in \mathbb{R}^d$, the submodularity ratio of $f_{\mathbf{z}}(\mathcal{S}) = -\mathbf{z}^\top \mathbf{V}_{\mathcal{S}}^{-1}\mathbf{z}$ satisfies $\gamma_k(f_{\mathbf{z}}) \geq \frac{1}{1+(k-1)\mu}$, where $\mu$ is the maximum coherence between any pair of elements in $\mathcal{X} \setminus \mathcal{S}$, given by $\mu = \max_{\mathbf{x}_i, \mathbf{x}_j \notin \mathcal{S}} |\mu_{i,j}|$, with $\mu_{i,j} = \frac{\mathbf{x}_i^\top \mathbf{V}_{\mathcal{S}}^{-1} \mathbf{x}_j}{\sqrt{1+\mathbf{x}_i^\top \mathbf{V}_{\mathcal{S}}^{-1}\mathbf{x}_i}\sqrt{1+\mathbf{x}_j^\top \mathbf{V}_{\mathcal{S}}^{-1}\mathbf{x}_j}}$.*

Lemma 1 gives a query-independent lower bound on the submodularity ratio of $f_{\mathbf{z}}$. In practice, for a given query $\mathbf{z}$, sets $\mathcal{S}$ and $\mathcal{L} = \{\mathbf{x}_1, \ldots, \mathbf{x}_k\} \subset \mathcal{X} \setminus \mathcal{S}$, one can compute the submodularity ratio (with abuse of notation) as $\gamma_k(f, \mathbf{z}, \mathcal{L}, \mathcal{S}) = \frac{\sum_{i=1}^{k} \Delta_i}{\sum_{i=1}^{k} \Delta_i - \sum_{i \neq j} \sqrt{\Delta_i \Delta_j} \mu_{i,j}}$, where $\Delta_i = \frac{(\mathbf{z}^\top \mathbf{V}_{\mathcal{S}}^{-1}\mathbf{x}_i)^2}{1+\mathbf{x}_i^\top \mathbf{V}_{\mathcal{S}}^{-1}\mathbf{x}_i}$ denotes the marginal gain in $f_{\mathbf{z}}$ of adding $\mathbf{x}_i$ to $\mathcal{S}$. The lower bound in Lemma 1 is derived by bounding this ratio using the Cauchy-Schwartz inequality and the maximum coherence $\mu$. Detailed proof of Lemma 1 is deferred to the appendix. The lower bound is empirically verified for all the datasets used in our experiments.

### 4.1 GREEDY ALGORITHM

We are now ready to present our Greedy algorithm for in-context example selection. Using the Sherman-Morrison formula, for any $\mathcal{S}' \subset \mathcal{S}$ such that $\mathcal{S} \setminus \mathcal{S}' = \{\mathbf{x}\}$, we get

$$f_{\mathbf{z}}(\mathcal{S}) = f_{\mathbf{z}}(\mathcal{S}') + \frac{(\mathbf{z}^\top \mathbf{V}_{\mathcal{S}'}^{-1}\mathbf{x})^2}{1 + \mathbf{x}^\top \mathbf{V}_{\mathcal{S}'}^{-1}\mathbf{x}} \,. \tag{5}$$

This naturally yields a greedy algorithm. We start with an emty set $\mathcal{S}_0 = \emptyset$ and at each step $i \in \{1, 2, \ldots, k\}$, we select the example that provides the maximum marginal gain over already selected examples $\mathcal{S}_{i-1} = \{\mathbf{x}_1, \ldots, \mathbf{x}_{i-1}\}$:

$$\mathbf{x}_i = \operatorname*{argmax}_{\mathbf{x} \in \mathcal{X} \setminus \mathcal{S}_{i-1}} \frac{(\mathbf{z}^\top \mathbf{V}_{\mathcal{S}_{i-1}}^{-1}\mathbf{x})^2}{1 + \mathbf{x}^\top \mathbf{V}_{\mathcal{S}_{i-1}}^{-1}\mathbf{x}} \,. \tag{6}$$

Although equation 6 involves inverting matrices $\mathbf{V}_i$ of dimension $d$, the rank-one nature of their sequential updates $\mathbf{V}_{\mathcal{S}_i} = \mathbf{V}_{\mathcal{S}_{i-1}} + \mathbf{x}_i \mathbf{x}_i^\top$ helps compute their inverse efficiently, again using the Sherman-Morrison formula:

$$\mathbf{V}_{\mathcal{S}_i}^{-1} = \mathbf{V}_{\mathcal{S}_{i-1}}^{-1} - \frac{\mathbf{V}_{\mathcal{S}_{i-1}}^{-1} \mathbf{x}_i \mathbf{x}_i^\top \mathbf{V}_{\mathcal{S}_{i-1}}^{-1}}{1 + \mathbf{x}_i^\top \mathbf{V}_{\mathcal{S}_{i-1}}^{-1} \mathbf{x}_i} \ . \tag{7}$$

This allows us to maintain the inverse matrix in $O(d^2)$ time per iteration, rather than recomputing it from scratch.

The following adaptation of a result from Das & Kempe (2011) using our lower bound on the submodularity ratio (Lemma 1) states the performance of the greedy algorithm under approximate submodularity.

**Theorem 1** (Greedy Algorithm Performance with Approximate Submodularity). *For any $\mathbf{z} \in \mathbb{R}^d$, let the greedy algorithm return a set $\mathcal{S}_{greedy}$ for the optimization problem in equation 4. Then*

$$f_{\mathbf{z}}(\mathcal{S}_{greedy}) \geq \left(1 - e^{-\frac{1}{1+(k-1)\mu}}\right) \cdot f(\mathcal{S}^*),$$

*where $S^*$ is an optimal solution of size at most $k$.*

Thus, the approximation guarantee degrades gracefully with the submodularity ratio $\gamma_k(f_{\mathbf{z}})$. When the submodularity ratio $\gamma_k(f_{\mathbf{z}})$ is close to 1 (i.e., when $k = 1$ or $\mu \approx 0$), greedy algorithms perform nearly as well as optimal algorithms (upto an $(1 - 1/e)$ factor). In practice, the value of $\gamma$ can be estimated, providing useful certificates of near-optimality for solutions obtained via greedy selection.

**Remark 1.** *For functions of the form $f_{\mathbf{z}}(\mathcal{S}) = -\mathbf{z}^\top \mathbf{V}_{\mathcal{S}}^{-1} \mathbf{z}$, where $\mathbf{V}_{\mathcal{S}} = \sum_{i \in \mathcal{S}} \mathbf{x}_i \mathbf{x}_i^\top + \beta \mathbf{I}_d$, increasing the regularization strength $\beta$ tends to make $f_{\mathbf{z}}(\mathcal{S})$ nearly submodular—that is, $\gamma_k(f) \to 1$. This shift further enhances the performance of the greedy algorithm in practice.*

**Selecting Diverse Examples** Equation 6 seeks the example $\mathbf{x}_i$ which is most relevant to the input query $\mathbf{z}$ in the geometry induced by the matrix $\mathbf{V}_{\mathcal{S}_{i-1}}^{-1}$. However, for in-context learning, both relevance (i.e., choosing examples similar to the input) and diversity (i.e., choosing similar examples) are essential, and need to be carefully balanced.

To select diverse examples, we seek the set $\mathcal{S}$ that fetches the maximum information about the unknown parameter vector $\theta \in \mathbb{R}^d$ from noisy linear responses of the form equation 1. To this end, let $\mathcal{S} \subset \mathcal{X}$ be a subset of size $k$. For a parameter vector $\theta \sim \mathcal{N}(\mathbf{0}_d, \beta \mathbf{I}_d)$, a design matrix $\mathbf{X}_{\mathcal{S}} \in \mathbb{R}^{k \times d}$ formed from the elements of $\mathcal{S}$, and response vector $\mathbf{y}_{\mathcal{S}} \in \mathbb{R}^k$ formed from responses $y = \mathbf{x}^\top \theta + \eta, \mathbf{x} \in \mathcal{S}$, with $\eta \sim \mathcal{N}(0, \beta)$, the information gain about $\theta$ from $\mathbf{y}_{\mathcal{S}}$ is given by the mutual information between $\theta$ and $\mathbf{y}_{\mathcal{S}}$, i.e., $I(\theta; \mathbf{y}_{\mathcal{S}}) := H(\mathbf{y}_{\mathcal{S}}) - H(\mathbf{y}_{\mathcal{S}}|\theta)$, where $H(\cdot)$ denotes the Shannon entropy (Cover, 1999). Since $\mathbf{y}_{\mathcal{S}}|\theta \sim \mathcal{N}(\mathbf{0}_k, \beta \mathbf{I}_k)$ and $\mathbf{y}_{\mathcal{S}} \sim \mathcal{N}(\mathbf{0}_k, \mathbf{X}_{\mathcal{S}}^\top \mathbf{X}_{\mathcal{S}} + \beta \mathbf{I}_k)$, information gain simplifies to

$$I(\theta; \mathbf{y}_{\mathcal{S}}) = \frac{1}{2} \log \det \left( \mathbf{I}_d + \frac{1}{\beta} \mathbf{X}_{\mathcal{S}}^\top \mathbf{X}_{\mathcal{S}} \right) ,$$

which follows from the fact that the entropy of a Gaussian distribution with covariance matrix $\Sigma$ is $\frac{1}{2} \log \det(2\pi e \Sigma)$. Therefore, to find the most informative (or, equivalently, diverse) set, we find the set with the maximum information gain. This, along with the fact that $\mathbf{V}_{\mathcal{S}} = \mathbf{X}_{\mathcal{S}}^\top \mathbf{X}_{\mathcal{S}} + \beta \mathbf{I}_d$, leads to the following optimization problem:

$$\max_{\mathcal{S} \subset \mathcal{X}, |\mathcal{S}| \leq k} g(\mathcal{S}) , \quad \text{where } g(\mathcal{S}) = \log \det(\mathbf{V}_{\mathcal{S}}) . \tag{8}$$

The function $g$ is known as the D-optimal design (Pukelsheim, 2006).

**Lemma 2** (Submodularity of $g$). *For any $\beta > 0$, the function $g(\mathcal{S}) = \log \det(\mathbf{V}_{\mathcal{S}})$ is monotone and submodular.*

The result holds from the matrix-determinant lemma, which, for any $\mathbf{x} \notin \mathcal{S}$, yields

$$\log \det(\mathbf{V}_{\mathcal{S}} + \mathbf{x}\mathbf{x}^\top) - \log \det(\mathbf{V}_{\mathcal{S}}) = \log(1 + \mathbf{x}^\top \mathbf{V}_{\mathcal{S}}^{-1} \mathbf{x}) .$$

Monotonicity follows since this increment $\log(1 + \mathbf{x}^\top \mathbf{V}_{\mathcal{S}}^{-1} \mathbf{x})$ is positive. Submodularity holds since this increment becomes smaller as $\mathbf{V}_{\mathcal{S}}$ grows (i.e., as the set $\mathcal{S}$ grows), due to redundancy in the directions already spanned. This also admits a natural greedy selection rule:

$$\mathbf{x}_i = \underset{\mathbf{x} \in \mathcal{X} \setminus \mathcal{S}_{i-1}}{\arg\max} \log(1 + \mathbf{x}^\top \mathbf{V}_{\mathcal{S}_{i-1}}^{-1} \mathbf{x}) , \tag{9}$$

which can also be computed efficiently using equation 7.

**Combined Relevance and Diversity** To select relevant as well as diverse examples, we maximize the combined objective $f_{\mathbf{z}}(\mathcal{S}) + g(\mathcal{S})$, which maintains both monotonicity and approximate submodularity. Therefore, we combine the corresponding greedy selection rules equation 6 and equation 9 to obtain

$$\mathbf{x}_i = \operatorname*{argmax}_{\mathbf{x} \in \mathcal{X} \setminus \mathcal{S}_{i-1}} \frac{(\mathbf{z}^\top \mathbf{V}_{\mathcal{S}_{i-1}}^{-1} \mathbf{x})^2}{1 + \mathbf{x}^\top \mathbf{V}_{\mathcal{S}_{i-1}}^{-1} \mathbf{x}} + \lambda \log(1 + \mathbf{x}^\top \mathbf{V}_{\mathcal{S}_{i-1}}^{-1} \mathbf{x}) \,,$$

where $\lambda \geq 0$ controls the trade-off between relevance and diversity. The set $\mathcal{S}$ returned by this greedy algorithm maintains the approximation guarantee of Theorem 1. We call this algorithm **L**inear **I**nformation **T**heoretic **E**xemplars (LITE) for ICL and present the pseudo-code in Algorithm 1.

### 4.2 MOVING BEYOND LINEARITY: KERNEL TRICK

In this section, we generalize our approach to the practical setting of non-linear models by assuming that conditioned on a test query $\mathbf{z} \in \mathbb{R}^d$, the response $y$ for any feature vector $\mathbf{x} \in \mathbb{R}^d$ is generated as $y = h(\mathbf{x}) + \eta$, where $\eta \sim \mathcal{N}(0, 1)$ and $h$ is an unknown element of a reproducing kernel Hilbert space (RKHS, see Schölkopf & Smola (2002)). An RKHS, denoted by $\mathcal{H}_k$, is completely characterized by a symmetric and positive semi-definite kernel $k : \mathbb{R}^d \times \mathbb{R}^d \to \mathbb{R}$ and vice-versa. Two commonly used kernels are the polynomial kernel $k_{\text{Poly}}(\mathbf{x}, \mathbf{x}') = (\mathbf{x}^\top \mathbf{x}' + c)^m$ and the Gaussian kernel $k_{\text{Gauss}}(\mathbf{x}, \mathbf{x}') = \exp\left(-\frac{\|\mathbf{x} - \mathbf{x}'\|^2}{2\sigma^2}\right)$, where $c, m, \sigma$ are hyperparameters of the kernels. The kernel trick says that there exists a feature map $\phi : \mathbb{R}^d \to \mathcal{H}_k$ such that $k(\mathbf{x}, \mathbf{x}') = \phi(\mathbf{x})^\top \phi(\mathbf{x}')$, where the inner product is associated with $\mathcal{H}_k$. This trick helps us to perform all computations in the high (possibly infinite) dimensional Hilbert space $\mathcal{H}_k$ analogously to those in the Euclidean space, but without the need for computing the inner product explicitly.

---

**Algorithm 1:** LINEAR AND INFORMATION THEORETIC EXEMPLARS (LITE)

**Require:** Dataset $\mathcal{X} = \{(\mathbf{x}_i, y_i)\}_{i=1}^n$; test query $\mathbf{z}$; subset size $k$; regularization $\beta > 0$; diversity weight $\lambda > 0$.

1: Initialize $\mathbf{V}^{-1} \leftarrow \frac{1}{\beta} \mathbf{I}_d$
2: Initialize available datapoints $\mathbf{mask} \leftarrow \mathbf{1}_n$
3: Initialize selected set $\mathcal{S} \leftarrow \emptyset$
4: **for** $j = 1$ to $k$ **do**
5:     $\mathcal{X}_{\text{available}} \leftarrow \{\mathbf{x}_i : \mathbf{mask}_i = 1\}$
6:     **for** each $\mathbf{x} \in \mathcal{X}_{\text{available}}$ **do**
7:         $\mathbf{p} \leftarrow \mathbf{V}^{-1}\mathbf{z}$; $\mathbf{u} \leftarrow \mathbf{V}^{-1}\mathbf{x}$
8:         $\text{rel}(\mathbf{x}) \leftarrow \frac{(\mathbf{x}^\top \mathbf{p})^2}{1 + \mathbf{x}^\top \mathbf{u}}$; $\text{div}(\mathbf{x}) \leftarrow \log(1 + \mathbf{x}^\top \mathbf{u})$
9:         $\text{score}(\mathbf{x}) \leftarrow \text{rel}(\mathbf{x}) + \lambda \cdot \text{div}(\mathbf{x})$
10:     **end for**
11:     $\mathbf{x}^* \leftarrow \operatorname{argmax}_{\mathbf{x} \in \mathcal{X}_{\text{available}}} \text{score}(\mathbf{x})$
12:     $\mathbf{u} \leftarrow \mathbf{V}^{-1}\mathbf{x}^*$; $\mathbf{V}^{-1} \leftarrow \mathbf{V}^{-1} - \frac{\mathbf{u}\mathbf{u}^\top}{1 + (\mathbf{x}^*)^\top \mathbf{u}}$
13:     $\mathcal{S} \leftarrow \mathcal{S} \cup \{\mathbf{x}^*\}$, update $\mathbf{mask}$ to exclude $\mathbf{x}^*$
14: **end for**
15: **return** $\mathcal{S}$

---

To see this, the greedy selection rule in the Euclidean space $\mathbb{R}^d$ can be lifted to the RKHS as

$$\mathbf{x}_i = \operatorname*{argmax}_{\mathbf{x} \in \mathcal{X} \setminus \mathcal{S}_{i-1}} \frac{\left(\phi(\mathbf{z})^\top \mathbf{V}_{\phi, \mathcal{S}_{i-1}}^{-1} \phi(\mathbf{x})\right)^2}{1 + \phi(\mathbf{x})^\top \mathbf{V}_{\phi, \mathcal{S}_{i-1}}^{-1} \phi(\mathbf{x})} + \lambda \log\left(1 + \phi(\mathbf{x})^\top \mathbf{V}_{\phi, \mathcal{S}_{i-1}}^{-1} \phi(\mathbf{x})\right) \,, \tag{10}$$

where $\mathbf{V}_{\phi, \mathcal{S}} = \beta \mathbf{I} + \sum_{i \in \mathcal{S}} \phi(\mathbf{x}_i) \phi(\mathbf{x}_i)^\top = \beta \mathbf{I} + \Phi_\mathcal{S}^\top \Phi_\mathcal{S}$ is the design covariance matrix in $\mathcal{H}_k$. Forming a design matrix $\Phi_\mathcal{S}$ from $\phi(\mathbf{x})$, $\mathbf{x} \in \mathcal{S}$ and using kernel trick, we obtain

$$\phi(\mathbf{x})^\top \mathbf{V}_{\phi, \mathcal{S}}^{-1} \phi(\mathbf{x}) = \phi(\mathbf{x})^\top (\Phi_\mathcal{S}^\top \Phi_\mathcal{S} + \beta \mathbf{I})^{-1} \phi(\mathbf{x})$$

$$= \frac{1}{\beta} \left(\phi(\mathbf{x})^\top \phi(\mathbf{x}) - \phi(\mathbf{x})^\top \Phi_\mathcal{S}^\top (\Phi_\mathcal{S} \Phi_\mathcal{S}^\top + \beta \mathbf{I})^{-1} \Phi_\mathcal{S} \phi(\mathbf{x})\right)$$

$$= \frac{1}{\beta} \left(k(\mathbf{x}, \mathbf{x}) - \mathbf{k}_\mathcal{S}(\mathbf{x})^\top (\mathbf{K}_\mathcal{S} + \beta \mathbf{I})^{-1} \mathbf{k}_\mathcal{S}(\mathbf{x})\right) \,, \tag{11}$$

where $\mathbf{K}_\mathcal{S} = [k(\mathbf{x}_i, \mathbf{x}_j)]_{\mathbf{x}_i, \mathbf{x}_j \in \mathcal{S}}$ is the gram matrix of size $|\mathcal{S}| \times |\mathcal{S}|$ and $\mathbf{k}_\mathcal{S}(x) = [k(\mathbf{x}, \mathbf{x}_i)]_{\mathbf{x}_i \in \mathcal{S}}$ is a vector of size $|\mathcal{S}|$.

Equation 11 helps the selection rule equation 10 to be implemented with only the kernel computations, avoiding matrix inverses and matrix-vector multiplications in (possibly) infinite-dimensional RKHS. This leads to our framework for in-context example selection, summarized below:

Note that KITE is a strict generalization of LITE (Algorithm 1) and captures it as a special case when the kernel is linear, i.e., when $k_{\text{lin}}(\mathbf{x}, \mathbf{x}') = \mathbf{x}^\top \mathbf{x}'$. In our experiments, we keep the kernel as a tunable hyperparameter of KITE.

> **KITE: Kernelized Information Theoretic Exemplars**
>
> At each step $i \in \{1, 2, \ldots, k\}$, select the example:
>
> $$\mathbf{x}_i = \operatorname*{argmax}_{\mathbf{x} \in \mathcal{X} \setminus \mathcal{S}_{i-1}} \frac{\left(k_{\mathcal{S}_{i-1}}(\mathbf{z}, \mathbf{x})\right)^2}{\beta + k_{\mathcal{S}_{i-1}}(\mathbf{x}, \mathbf{x})} + \lambda \log \left(\beta + k_{\mathcal{S}_{i-1}}(\mathbf{x}, \mathbf{x})\right),$$
>
> where $k_{\mathcal{S}}(\mathbf{z}, \mathbf{x}) = k(\mathbf{z}, \mathbf{x}) - \mathbf{k}_{\mathcal{S}}(\mathbf{z})^\top (\mathbf{K}_{\mathcal{S}} + \beta \mathbf{I})^{-1} \mathbf{k}_{\mathcal{S}}(\mathbf{x})$.

## 5 EXPERIMENTS

We present a comprehensive empirical analysis of our proposed approach, evaluating its performance across a variety of open-source datasets and models.

| | SST-2 | | | SST-5 | | | CMSQA | | | MRPC | | | QNLI | | | HellaSwag | | |
|---|---|---|---|---|---|---|---|---|---|---|---|---|---|---|---|---|---|---|
| **Method** | GN | QW | LL | GN | QW | LL | GN | QW | LL | GN | QW | LL | GN | QW | LL | GN | QW | LL |
| Random | 86.32 | 63.36 | 89.84 | 32.31 | 40.53 | 42.41 | 42.25 | 65.90 | 67.22 | 66.38 | 70.45 | 71.12 | 57.56 | 70.21 | 68.43 | 41.75 | 66.30 | 65.21 |
| BM25 | 90.14 | 67.78 | 91.63 | 36.05 | 47.86 | **47.85** | 42.91 | 70.02 | 70.92 | 66.96 | 73.77 | 70.58 | 62.12 | **71.11** | 69.38 | 41.34 | 67.85 | 66.32 |
| Dense | 88.99 | 72.94 | 92.55 | 35.96 | 46.64 | 44.14 | 43.98 | 70.18 | 72.23 | 65.83 | 73.92 | 71.81 | 64.42 | 70.73 | 70.04 | 40.44 | 68.80 | 67.88 |
| DPP | 90.25 | 71.22 | 91.86 | 36.84 | 47.88 | 46.45 | 37.53 | 70.05 | 71.65 | 68.04 | 74.25 | 70.63 | 63.62 | 70.53 | 70.08 | 40.56 | 69.32 | 68.55 |
| KITE | **93.35** | **74.41** | **94.28** | **40.60** | **49.59** | 47.59 | **46.60** | **71.25** | **72.89** | **68.38** | **75.27** | **71.98** | **65.32** | 71.05 | **71.68** | **41.68** | **71.02** | **70.12** |

Table 1: **Evaluation Results.** We compare classification accuracy (%) of KITE against retrieval baselines on six few-shot benchmarks. Model abbreviations: GN (GPT-Neo 2.7B), QW (Qwen 2.5–1.5B), LL (Llama 3.2–3B).

**Implementation Details.** We conduct our evaluations using three open-source, state-of-the-art language models: GPT-Neo-2.7B (Black et al., 2021), Qwen 2.5-1.5B (Hui et al., 2024), and Llama-3.2-3B (Grattafiori et al., 2024). For all baselines, we fix the number of in-context examples to 50 during inference, truncating as necessary based on the model's maximum context length. We represent textual inputs using embeddings obtained from `bert-base-uncased` (Devlin et al., 2019). We experiment with three kernels: (i) Linear kernel $k_{\text{lin}}(\mathbf{x}, \mathbf{x}')$ (which reduces to LITE; see Algorithm 1), (ii) Polynomial kernel $k_{\text{poly}}(\mathbf{x}, \mathbf{x}')$ with varying degree $m$ and (iii) Gaussian kernel $k_{\text{Gauss}}(\mathbf{x}, \mathbf{x}')$ (length-scale $\sigma = 1.0$). We use a regularization parameter of $\beta = 0.02$ and a diversity parameter of $\lambda = 0.5$. For KITE in Table 1, we report the best result across the three kernel choices. A detailed ablation study on the impact of the kernel choice is presented in Table 3. We defer the mathematical formulation for each kernel to Appendix.

**Datasets.** To empirically demonstrate the efficacy of KITE, we evaluate on five few-shot classification datasets: SST-2 & 5 (Socher et al., 2013) (single-sentence sentiment), CMSQA (Talmor et al., 2019) (question answering), MRPC (Dolan et al., 2004) (sentence-pair paraphrase), QNLI (Wang et al., 2018) (binary entailment classification), and HellaSwag (Zellers et al., 2019) (common sense NLI).

For each task, we use the corresponding validation split for evaluation and employ the deduplicated training split as the candidate exemplar set.

**Evaluation Metrics.** Following prior work (Brown et al., 2020a), we formulate all classification tasks as multiple-choice problems. we construct input prompts by concatenating the given context with each candidate label (i.e., "context + label"). We then compute the conditional log-likelihood of generating the label tokens given this combined input prompt, and select the candidate with the highest log-likelihood as the model's prediction. Model performance is evaluated using accuracy, defined as the fraction of correctly predicted examples on the validation split.

**Baselines.** For a fair and comprehensive evaluation, we compare KITE against several baselines, including Random, BM25 (Robertson et al., 2009), top-$k$ with Dense embddings (Dense) (Liu et al., 2021), and DPP-based retrieval strategies (Ye et al., 2023a).

**Results.** We report our evaluation results in Table 1. We find that KITE consistently outperforms all baselines across most datasets and model architectures. On GPT-Neo-2.7B, for example, KITE delivers an accuracy improvement of +4.55% on SST-5 (Socher et al., 2013) and +3.69% on CMSQA (Talmor et al., 2019) over BM25 (Robertson et al., 2009). Further, this trend is consistently observed across all evaluated models. With Qwen-2.5-1.5B, KITE surpasses the strongest baseline, DPP (Ye et al., 2023a), on four out of five datasets, achieving notable gains of +1.71% on SST-5 and +1.70% on HellaSwag. Similarly, on Llama-3B, KITE achieves the highest accuracy on four datasets, including a substantial +2.24% improvement on HellaSwag over the state-of-the-art (Ye et al., 2023a). Across all

| Dataset | Vary $k$ (fixed $\beta=1$) | | | | | | | Vary $\beta$ (fixed $k=20$) | | | | | |
|---|---|---|---|---|---|---|---|---|---|---|---|---|---|
| | 5 | 10 | 15 | 20 | 25 | 30 | 35 | 1 | 3 | 5 | 7 | 9 | 11 |
| SST–5 | 0.748 | 0.685 | 0.705 | 0.689 | 0.703 | 0.716 | 0.696 | 0.689 | 0.744 | 0.719 | 0.758 | 0.792 | 0.827 |
| CMSQA | 0.852 | 0.811 | 0.816 | 0.841 | 0.830 | 0.797 | 0.836 | 0.841 | 0.804 | 0.884 | 0.843 | 0.907 | 0.886 |
| MRPC | 0.876 | 0.840 | 0.793 | 0.816 | 0.850 | 0.887 | 0.848 | 0.816 | 0.765 | 0.803 | 0.824 | 0.824 | 0.898 |
| QNLI | 0.844 | 0.801 | 0.900 | 0.893 | 0.875 | 0.837 | 0.877 | 0.893 | 0.829 | 0.823 | 0.919 | 0.912 | 0.882 |
| HellaSwag | 0.395 | 0.601 | 0.525 | 0.596 | 0.839 | 0.908 | 0.806 | 0.596 | 0.803 | 0.767 | 0.806 | 0.935 | 0.852 |

Table 2: **Analysis on Submodularity ratio $\gamma_{\min}$. Left:** We vary $k$ while fixing $\beta=1$; **Right:** We vary $\beta$ while fixing $k=20$.

15 evaluation settings (five datasets and three models), KITE attains the highest accuracy in 13 cases, underscoring its robustness and efficacy as an exemplar retrieval framework.

**Ablation study on kernel function.** The choice of kernel is a critical hyperparameter in our KITE framework, as it defines the geometry of the feature space in which relevance and diversity are measured. To clearly understand its impact, we conducted an ablation study comparing the performance of three distinct kernels: Linear (which reduces KITE to the LITE algorithm), Polynomial (with degree $m=3$), and Gaussian RBF ($\sigma=1.0$). The results, detailed in Table 3, reveal that no single kernel is universally optimal; the best choice is contingent on the specific dataset and its underlying data distribution.

The strong performance of the non-linear kernels across most datasets validates our core motivation for moving beyond the linear model of LITE. It underscores the importance of the kernel trick in empowering the model to capture richer, non-linear relationships between exemplars, which is essential for effective in-context learning. The main results reported in Table 1 represent the best performance achieved across these kernels for each task, highlighting the consistent advantage conferred by a well-tuned kernelized approach.

**Empirical validation of submodularity.** We empirically validate the submodularity of our set function, $f_z(\mathcal{S}) = -\mathbf{z}^\top \mathbf{V}_{\mathcal{S}}^{-1} \mathbf{z}$ by estimating its submodularity ratio (Def. 2) on real-world text embeddings. For this analysis, we used the first 500 examples from each dataset to generate embeddings. We created two distinct types: demonstration embeddings, by concatenating an input with its gold label, and query embeddings, from the input alone.

| Dataset | Linear | Polynomial | Gaussian RBF |
|---|---|---|---|
| SST-5 | 47.95 | 48.38 | **49.59** |
| CMSQA | 69.86 | 71.06 | **71.25** |
| MRPC | **75.27** | 71.22 | 67.15 |
| QNLI | 70.46 | 70.38 | **71.05** |
| HellaSwag | 67.18 | **71.02** | 70.67 |

Table 3: **Ablation study on the kernel function.** We report accuracy for KITE using different kernels on Qwen-1.5B.

To estimate the ratio, we ran a Monte Carlo simulation across a grid of hyperparameters for subset size $k$ and regularization $\beta$. In each trial, we sampled a disjoint triplet $(\mathcal{S}, \mathcal{L}, \mathbf{z})$, where $\mathcal{S}$ is a random set of demonstration embeddings, $\mathcal{L}$ is a diverse set of up to $k$ demonstration embeddings selected via farthest-point sampling, and $\mathbf{z}$ is a query embedding. We then computed the ratio and recorded the minimum value observed ($\gamma_{\min}$) for each configuration.

As shown in Table 2, our experiments reveal a consistently high $\gamma_{\min}$ across all datasets and settings. These values, typically at or above 0.8, provide strong empirical evidence that our objective function is approximately submodular in practice. This finding justifies our use of a greedy algorithm, as it is expected to yield near-optimal results.

## 6 CONCLUSION

In this paper, we introduce KITE, a principled, information-theoretic framework that treats in-context exemplar selection as a query-specific optimization problem. By leveraging an approximately submodular objective and the kernel trick, our algorithm efficiently selects exemplars that balance relevance and diversity, an approach supported by theoretical performance guarantees. Extensive experiments demonstrate that KITE consistently outperforms strong retrieval baselines across multiple classification datasets and language models, validating its efficacy and effectiveness for in-context example selection. Extending KITE to generative tasks, which involve producing multiple dependent output tokens, is an important direction for future work.

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
