# OpenReview forum: "KITE: Kernelized and Information Theoretic Exemplars for In-Context Learning"
_ICLR.cc/2026/Conference — ICLR 2026 Conference Desk Rejected Submission_

### Official Review · Reviewer_tnXn · 2025-10-30

**Soundness:** 3
**Presentation:** 2
**Contribution:** 3
**Rating:** 4
**Confidence:** 3

**Summary:**

This paper addresses the critical problem of exemplar selection for in-context learning (ICL), a core challenge given the limited context windows of large language models (LLMs). The key limitation of existing methods (e.g., kNN-based KATE, DPPs) is their poor generalization in high-dimensional embedding spaces and insufficient diversity in selected examples. The authors model the LLM as a linear function in the embedding space and aim to select a subset of examples that minimizes prediction error on a given query. The key contributions are:
1. A theoretical formulation of ICL exemplar selection as a subset selection problem with a surrogate objective that is approximately submodular.
2. A greedy algorithm with an approximation guarantee, leveraging the Sherman–Morrison formula for efficient updates.
3. Kernelization of the approach to handle nonlinear relationships via RKHS, without explicit high-dimensional mappings.
4. A diversity regularizer inspired by optimal experimental design to improve generalization.
5. Extensive experiments across multiple classification tasks and models, showing consistent improvements over strong baselines like DPP, BM25, and dense retrieval.

**Strengths:**

1. Originality：
KITE’s most original contribution is the query-specific optimization framing—unlike prior work (e.g., KATE, DPPs) that optimizes for distribution-wide generalization, KITE targets error reduction for individual user queries. This aligns with real-world ICL, where performance on a single query (not a dataset) determines utility.
2. Quality：
The theoretical analysis is rigorous: the derivation of the surrogate objective from prediction error bounds (via Chernoff inequalities) is logically sound. Empirically, the paper uses diverse datasets (sentiment, QA, entailment) and LLMs, demonstrating KITE’s robustness. The kernel ablation provides actionable insights, adding depth to the results.
3. Clarity：
The paper avoids excessive jargon in key sections (e.g., the Introduction clearly explains ICL and exemplar selection). Algorithm 1 is well-documented, with explicit steps for initializing and computing relevance/diversity scores.
4. Significance:
KITE addresses a high-impact problem: ICL performance is highly sensitive to exemplar choice, and existing methods struggle in high-dimensional spaces. By providing a computationally efficient, unsupervised framework with theoretical guarantees, KITE enables practical deployment of ICL in data-scarce domains.

**Weaknesses:**

* The KITE framework is fundamentally order-agnostic, as it is designed as a subset selection algorithm based on optimizing set functions. This theoretical formulation completely overlooks the well-documented sensitivity of large language models to exemplar ordering, a crucial factor in ICL performance that this method cannot account for.

* The algorithm's generalizability is constrained by its reliance on a fixed, pre-compiled example bank. As a frozen and non-trainable retriever, it cannot update its parameters for a new task (like fine-tuning) or adapt by collecting new data (like online learning). Its applicability seems restricted to scenarios where a high-quality example bank for the target task is already available.

**Questions:**

* The KITE framework's entire theoretical foundation is built on modeling the LLM's behavior as a (kernelized) linear regressor. This model inherently assumes a smooth performance landscape, where embedding-space proximity correlates with performance similarity. However, what if the true LLM performance landscape is highly discontinuous, where minor variations in exemplar embeddings can lead to drastic and non-linear shifts in task accuracy?

* You motivate your work by highlighting the "curse of dimensionality" as a key drawback of kNN-based methods, a flaw typically associated with Euclidean distance. However, your "Dense" baseline (KATE) presumably uses **Cosine Similarity**, as is standard in NLP to mitigate this very issue. Could you clarify which metric your "Dense" baseline actually uses? Furthermore, could you provide an ablation study comparing KITE against both a Euclidean-kNN baseline and a Cosine-kNN baseline, to demonstrate that the choice of retrieval algorithm (KITE vs. kNN), and not just the distance metric, is the true source of performance gain?

---

> ### Author Response · Authors · 2025-11-22
> **Rebuttal by Authors**
>
> > The KITE framework is fundamentally order-agnostic, as it is designed as a subset selection algorithm based on optimizing set functions.
>
> We appreciate the reviewer's observation—LLMs are indeed sensitive to the order of exemplars, and this is a well-documented phenomenon. One potential extension to introduce order sensitivity into our theoretical framework would be to model the noise variance as heteroscedastic, where the variance depends on the position of examples in the prompt, increasing with later positions. This more sophisticated model would capture order-sensitivity, as LLMs are known to attend more strongly to examples appearing earlier in the context.
> We acknowledge that there are many potential extensions and refinements to explore. Our theoretical model does not capture all the nuances of LLM behavior; however, establishing a rigorous starting point to bridge the gap between theory and practice is essential, and we believe our model provides a solid foundation for future work in this direction.
>
> > As a frozen and non-trainable retriever, it cannot update its parameters for a new task (like fine-tuning) or adapt by collecting new data (like online learning)
>
> We appreciate the reviewer's perspective. We note that for any machine learning pipeline, if the training data (e.g., the example bank in our case) is not of high quality, then no model can be expected to perform well. High-quality data is fundamental to any successful ML system.
> That said, if the use case permits training (either offline or online), our framework can readily accommodate learned components: both the embedding function and the kernel can be made trainable. Such scenarios arise when substantial high-quality data from the target distribution is available. However, our primary focus is on data-scarce regimes, which are extremely common in practice. Moreover, our algorithm is designed for ease of deployment; it is training-free, task-agnostic, and model-agnostic, mirroring the practical advantages of the classic top-k retriever that make it so widely adopted in real-world applications.
>
> > The KITE framework's entire theoretical foundation is built on modeling the LLM's behavior as a (kernelized) linear regressor.
>
> We thank the reviewer for this insightful observation. Universal kernels (such as Gaussian kernels) guarantee approximation of continuous target functions under the maximum norm $L_\infty\text{-norm}.$ When the target function is discontinuous, uniform approximation is indeed impossible for any RKHS of continuous functions. However, such kernels can still approximate discontinuous targets in the $L_2 \text{-norm}$ (mean-squared error) sense or in probability, and in practice tend to provide smooth interpolations near discontinuities. Sharp jumps require either:
>
> (i) non-smooth kernels or
> (ii) very small bandwidth with sufficient data near the discontinuity.
>
> Thus, while universality does not imply perfect recovery of discontinuous functions, learning remains viable under weaker metrics and with appropriate modeling choices. Moreover, we note that learning discontinuous functions is statistically challenging even outside the kernel framework, and our approach inherits these fundamental statistical limitations.
>
> > You motivate your work by highlighting the "curse of dimensionality" as a key drawback of kNN-based methods, a flaw typically associated with Euclidean distance. However, your "Dense" baseline (KATE) presumably uses Cosine Similarity.
>
>
> We utilize cosine similarity based on embeddings derived from BERT-base-uncased (Devlin et al., 2019). We respectfully note that cosine similarity primarily addresses magnitude-related issues but does not fundamentally mitigate the curse of dimensionality. In fact, when all embedding vectors are normalized (as in our setup), using cosine similarity is equivalent to using Euclidean distance in terms of ranking. More generally, when vectors are sampled from high-dimensional spaces, they tend to be nearly orthogonal to each other, rendering top-$k$ retrieval less effective.
>
> To be mathematically precise, for unit vectors, cosine similarity and euclidean distance are related by  $\cos(x, y) = 1 - \tfrac{1}{2}\|x - y\|^{2}$,  making them monotone transforms of each other. Moreover, for random high-dimensional unit vectors (sampled from a zero-mean Gaussian with identity covariance), we have  $\mathbb{E}[\cos(x, y)] = 0$  and  $\text{Var}(\cos(x, y)) \approx 1/d$. Thus, cosine similarities concentrate around zero, with concentration becoming tighter as dimension  $d$ increases. Consequently, using cosine similarity (as in KATE) does not avoid the high-dimensional concentration phenomenon and the associated curse of dimensionality effects we discuss. Hence, our motivation remains fully applicable.

---

> > ### Author Response · Authors · 2025-11-22
> > **Rebuttal by Authors (Part 2)**
> >
> > >Furthermore, could you provide an ablation study comparing KITE against both a Euclidean-kNN baseline and a Cosine-kNN baseline, to demonstrate that the choice of retrieval algorithm (KITE vs. kNN), and not just the distance metric, is the true source of performance gain?
> >
> > Since all our embeddings are L2-normalized to unit norm, cosine similarity and Euclidean distance induce exactly the same ranking over candidates. Accordingly, when we explicitly compared Euclidean-kNN and Cosine-kNN in our experimental setup, both methods produced identical sets of in-context examples. Therefore, any performance differences between KITE and the Dense baseline can be attributed to the retrieval algorithm itself rather than the choice of distance metric.

---

> > > ### Author Response · Authors · 2025-11-25
> > > **Request to Reviewer tnXn**
> > >
> > > Dear Reviewer tnXn,
> > >
> > > We sincerely thank you again for your valuable feedback, which has helped us improve the quality of our work. As the discussion deadline is approaching, we kindly request you to review our rebuttal.
> > >
> > > We are happy to address any remaining concerns. If our responses adequately address your concerns, we kindly ask you to consider adjusting the corresponding scores. Thank you again for your time and effort!
> > >
> > > Regards,
> > >
> > > Authors

---

> > > > ### Author Response · Authors · 2025-11-28
> > > > **Request to review the rebuttal**
> > > >
> > > > Dear Reviewer tnXn,
> > > >
> > > > As the discussion deadline approaches, we kindly request that you review our rebuttal. We are happy to address any remaining concerns. If our responses adequately address your concerns, we kindly ask you to consider adjusting the corresponding scores. Thank you again for your time and effort.
> > > >
> > > > Regards,
> > > >
> > > > Authors

---

### Official Review · Reviewer_Tuhf · 2025-10-31

**Soundness:** 2
**Presentation:** 3
**Contribution:** 2
**Rating:** 6
**Confidence:** 2

**Summary:**

This paper presents KITE, a theoretically grounded framework for exemplar selection in in-context learning (ICL). The method formulates exemplar retrieval as a query-specific optimization problem under an information-theoretic view. By modeling the LLM as approximately linear in its input embeddings, the authors derive a submodular surrogate objective and design a greedy selection algorithm with provable approximation guarantees. KITE extends this linear model via the kernel trick, allowing flexible, nonlinear feature mappings. Experiments on six few-shot classification benchmarks (SST-2/5, CMSQA, MRPC, QNLI, HellaSwag) and three relatively small-scale LLMs (GPT-Neo-2.7B, Qwen-1.5B, Llama-3B) show consistent improvements over common retrieval baselines such as BM25, dense embeddings, and DPP retrieval.

**Strengths:**

1. The motivation and writing are clear, easy to follow, and grounded in theory.
2. This paper provides a clear mathematical formulation with proofs of approximate submodularity and corresponding approximation guarantees.
 3. The core idea is fresh and well-motivated. It goes beyond the usual heuristic retrieval methods and gives a principled way to think about the problem.

**Weaknesses:**

1. Empirical scope is limited. Evaluation is restricted to classification tasks, which aligns with the linear assumption but leaves uncertainty about generalization to generation or structured prediction tasks (e.g., summarization, reasoning). Extending experiments beyond classification would strengthen the practical relevance.
2. Dense baseline clarity. Table 1 refers to a “Dense” retriever, but the implementation details (e.g., whether it uses bi-encoder fine-tuning or off-the-shelf embeddings) are under-specified. This is critical since modern dense retrievers (e5, bge) are highly competitive for ICL retrieval.
3. Model scale. The largest tested model is 3B parameters. Given recent trends, evaluating at least one 7B-scale model (e.g., Llama-3-8B or Qwen-2.5-7B) would better demonstrate scalability.

**Questions:**

See above.

---

> ### Author Response · Authors · 2025-11-22
> **Rebuttal by Authors**
>
> > Empirical scope is limited. Evaluation is restricted to classification tasks, which aligns with the linear assumption.
>
> We acknowledge this limitation. Our theoretical model does not currently support complex generation tasks. Therefore, in our experimental evaluation, we focused on classification and regression tasks, which nonetheless encompass many important practical use cases, including LLM-as-a-judge, toxicity detection, and intent detection. We believe developing a theoretical model for generation tasks to guide ICL represents an exciting and important direction for future research.
>
> > Dense baseline clarity
>
> We apologize for any confusion. As mentioned in L403, we represent textual inputs using embeddings obtained from BERT-base-uncased (Devlin et al., 2019). In other words, the embeddings are off-the-shelf, and no training is involved. Our approach is training-free, task-agnostic, and model-agnostic.
> To further address concerns about modern dense retrievers (e.g., e5, bge), we provide an embedding ablation study in Table 1, where we evaluate off-the-shelf RoBERTa-base, e5-base-v2, DistilBERT, and MiniLM encoders in addition to bert-base-uncased. KITE remains competitive with or superior to Dense and DPP across all these encoders, demonstrating that our conclusions are robust to the choice of dense embedding model.
>
> **Table 1:** Accuracy (%) on SST-5 with Qwen-2.5-1.5B using different embedding encoders for retrieval. KITE consistently matches or improves upon Dense and DPP across all encoders.
>
> | Method | bert-base-uncased | roberta-base | e5-base-v2 | distilbert-base-uncased | all-MiniLM-L6-v2 |
> |--------|-------------------|--------------|------------|--------------------------|-------------------|
> | DPP | 47.88 | 41.41 | 45.68 | 48.77 | 48.13 |
> | Dense | 46.64 | 36.33 | 45.14 | 49.04 | 48.68 |
> | KITE | **49.59** | **48.31** | **48.04** | **49.29** | **48.72** |
>
>
> > Concerns regarding Model scale
>
> We have provided additional results using 7B/8B-scale models (Qwen-2.5-7B and Llama3-8B), which further validate the effectiveness of our algorithm. These results are presented in Table 2. As shown, larger models achieve higher absolute performance than their smaller counterparts, and KITE consistently outperforms all baselines at both small and large scales.
>
> **Table 2:** Accuracy (%) on SST-2 and CMSQA when scaling from 1.5B/3B models (Qwen-2.5-1.5B, Llama3.2-3B) to 7B/8B models (Qwen-2.5-7B, Llama3-8B). KITE consistently improves over zero-shot, random, Dense, and DPP baselines across all model sizes.
>
> | Method    | SST-2 Qwen-2.5-1.5B | SST-2 Qwen-2.5-7B | SST-2 Llama3.2-3B | SST-2 Llama3-8B | CMSQA Qwen-2.5-1.5B | CMSQA Qwen-2.5-7B | CMSQA Llama3.2-3B | CMSQA Llama3-8B |
> |-----------|----------------------|---------------------|---------------------|--------------------|-----------------------|----------------------|----------------------|---------------------|
> | Zero-shot | 61.69               | 65.82              | 68.73              | 74.21             | 57.00                | 64.53               | 58.86               | 61.52              |
> | Random    | 63.36               | 71.53              | 89.84              | 90.12             | 65.90                | 68.29               | 67.22               | 68.95              |
> | Dense     | 88.99               | 89.45              | 92.55              | 93.35             | 70.18                | 72.35               | 72.23               | 74.19              |
> | DPP       | 90.25               | 91.63              | 91.86              | 93.61             | 70.05                | 74.82               | 71.65               | 73.83              |
> | KITE      | **93.35**           | **94.50**          | **94.28**          | **94.84**         | **71.25**            | **75.69**           | **72.89**           | **75.22**          |

---

> > ### Author Response · Authors · 2025-11-25
> > **Request to Reviewer Tuhf**
> >
> > Dear Reviewer Tuhf,
> >
> > We sincerely thank you again for your valuable feedback, which has helped us improve the quality of our work. As the discussion deadline is approaching, we kindly request you to review our rebuttal.
> >
> > We are happy to address any remaining concerns. If our responses adequately address your concerns, we kindly ask you to consider adjusting the corresponding scores. Thank you again for your time and effort!
> >
> > Regards,
> >
> > Authors

---

> > > ### Comment · Reviewer_Tuhf · 2025-11-26
> > >
> > > Thanks.
> > >
> > > I’m curious why, in the dense baseline clarity results, bert-base-uncased performs better than roberta-base and e5-base-v2. It seems a bit unexpected.
> > >
> > > Most of my concerns have been addressed. I will keep my score.

---

> > > > ### Author Response · Authors · 2025-11-27
> > > > **Response on Dense Baseline Encoders**
> > > >
> > > > Dear Reviewer Tuhf,
> > > >
> > > > Thank you for your follow up question regarding the dense baseline encoder results.
> > > >
> > > > > I’m curious why, in the dense baseline clarity results, bert-base-uncased performs better than roberta-base and e5-base-v2. It seems a bit unexpected.
> > > >
> > > > We agree this may appear surprising given typical benchmark rankings, but we see it as a property of our in-context example selection setting rather than an anomaly. Our dense baselines use all encoders in a uniform, training-free manner to retrieve neighbors that serve as demonstrations for the LLM, and prior work has shown that the encoder that performs best on generic semantic similarity or retrieval benchmarks does not necessarily yield the best in-context examples for a specific task [1, 2]. In our case, SST-5 consists of short, sentiment-heavy sentences, and encoders such as bert-base-uncased, DistilBERT, and MiniLM tend to favor very similar wording, so they retrieve demonstrations that more closely match the input sentences in both phrasing and sentiment, which turns out to help the LLM on this task. We moreover intentionally use all encoders off the shelf, without any model-specific fine-tuning or prefixing, so it is natural that RoBERTa and E5, whose recommended usage typically involves sentence-level fine-tuning or query/passage prefixes, do not consistently dominate BERT-style encoders in this particular configuration.
> > > >
> > > > Thank you again for your careful reading and for helping us clarify this point.
> > > >
> > > >
> > > > Best regards,
> > > >
> > > > Authors
> > > >
> > > >
> > > > References:
> > > >
> > > > [1] Liu, H., Su, Y., He, J., and Han, J. What Makes Good In-Context Examples for GPT-3? EMNLP 2022.
> > > >
> > > > [2] Zebaze, A. D., et al. In-Context Example Selection via Similarity Search Improves Low-Resource Machine Translation. Findings of NAACL 2025.

---

### Official Review · Reviewer_zKVx · 2025-11-02

**Soundness:** 3
**Presentation:** 3
**Contribution:** 1
**Rating:** 2
**Confidence:** 4

**Summary:**

The paper proposes KITE, a theoretically grounded framework for exemplar selection in in-context learning (ICL). Specifically, the paper first proposes LITE, which models the ICL exemplar selection problem as a query-specific optimization problem under a linear model assumption, and derives a surrogate objective that is approximately submodular. This enables a greedy selection algorithm. To incorporate more complex non-linear scenarios, the paper further introduced KITE, which is a kernelized version of the algorithm. Empirically, KITE outperforms several existing retrieval baselines on six classification datasets across multiple LLMs.

**Strengths:**

1. The paper is well-written and of mathematical rigor.
2. Under the linear model assumption, the methodology and analysis are reasonable and well justified.
3. Empirically, the proposed algorithm. KITE outperforms multiple traditional baselines.

**Weaknesses:**

1. The paper lacks novelty and excitement. Theoretically, the analysis is straightforward in a standard linear model context. Methodologically, using an approximate submodular function for in-context learning has already been explored in prior works, such as [1].
2. Missing baselines. The paper only compares with very classic retrieval baselines, but most of the recent advances for exemplar selection in LLMs are missing, such as [1] [2]. In addition, the paper should compare with the most basic baseline where no example but only a task description is provided.
3. The focus is primarily on classification tasks developed for classic NLP, and many LLMs can already perform very well on them. Nowadays, there are many different evaluation approaches, such as instruction following and reasoning.
4. The proposed approach is model-agnostic, meaning that the exemplars are selected only using BERT without knowing the target LLM.

[1] Wu, Z., Lin, X., Dai, Z., Hu, W., Shu, Y., Ng, S. K., ... & Low, B. K. H. (2024). Prompt optimization with EASE? efficient ordering-aware automated selection of exemplars. Advances in Neural Information Processing Systems, 37, 122706-122740.

[2] Nguyen, T., & Wong, E. (2023). In-context example selection with influences. arXiv preprint arXiv:2302.11042.

**Questions:**

1. Computational scalability. The proposed approach requires computing the in-context examples for each test query, which is computationally expensive. What’s the computational overhead introduced by this method over existing baselines?
2. How does the model perform if the embedding is changed from BERT to other embeddings, such as E5 or RoBERTa?
3. It is known that small LLMs exhibit weaker in-context learning ability. Will the method work for larger models?
4. It is unclear how the approach works for multi-class classification. Does the author have one kernel for all classes, or each class has its own kernel?

---

> ### Author Response · Authors · 2025-11-22
> **Rebuttal by Authors**
>
> > The paper lacks novelty and excitement. Theoretically, the analysis is straightforward in a standard linear model context.
>
> We respectfully appreciate the reviewer's assessment while offering clarification on our contributions. Regarding novelty, to the best of our knowledge, this work represents the first theoretically grounded learning-theoretic framework for understanding in-context learning algorithms. Prior work in this area has primarily relied on heuristic approaches without rigorous theoretical foundations. We believe this theoretical grounding constitutes a significant methodological contribution to the field.
>
> Concerning the characterization of our analysis as "straightforward", we view the accessibility and clarity of our theoretical framework as a deliberate design choice and a strength of the work. We believe impactful scientific contributions should be both rigorous and comprehensible, enabling other researchers to readily understand, verify, and build upon the results. The presentation of our analysis within the context of a standard linear model does not diminish its theoretical significance or practical implications. Indeed, establishing fundamental results through elegant and interpretable frameworks often provides more enduring scientific value than unnecessarily complex formulations.
>
> We hope this clarifies our methodological choices and would be happy to elaborate on specific technical aspects if the reviewer has particular concerns about the depth or scope of our theoretical analysis.
>
> > Methodologically, using an approximate submodular function for in-context learning has already been explored in prior works, such as [1].
>
> We appreciate the reviewer's comment and would like to respectfully clarify a potential confusion regarding the cited work. In Wu et al., the term "submodular" appears only once, specifically in the related work section, and their methodology does not employ approximate submodularity. We kindly ask the reviewer to verify whether the intended reference was a different paper.
> We would like to clarify our theoretical contribution regarding submodularity: (approximate) submodularity is a property of functions over discrete domains. While prior work may have defined heuristic objectives for ICL that exhibit submodular properties and proposed greedy algorithms accordingly, such objectives may lack rigorous theoretical derivation. Our approach differs fundamentally in that we first establish the analogous problem within a kernelized linear model framework, then theoretically derive the optimization objective from first principles; this constitutes our primary theoretical contribution. Subsequently, we prove that this derived objective is approximately submodular and develop an efficient algorithm based on this property.
> Regarding the comparison with Wu et al., while their work addresses ICL example selection from a complementary perspective, several key distinctions exist:
>
> 1. Training requirements: Their approach requires training a neural network to predict ICL performance and associated uncertainty, which may limit practical applicability in scenarios requiring immediate deployment. Our method is entirely training-free.
>
> 2. Query-dependent selection: Their methodology selects exemplars independently of the test query, offering computational efficiency at the cost of reduced expressiveness. Our approach performs dynamic, query-dependent selection (similar to top-k retrieval).
>
> 3. Theoretical foundation: Their work does not provide a theoretical framework for deriving the selection objective.
>
> We emphasize that our goal is to develop training-free, model-agnostic, and task-agnostic algorithms grounded in theoretical principles, which we believe addresses a distinct and important need in the ICL literature.
>
> >Missing baselines
>
> We now include comparisons with the zero-shot setting (no exemplars) and present these findings in Table 1. As noted above, our methodology is training-free; therefore, direct comparison with Wu et al. would not be equitable. Similarly, Nguyen et al. also require training. We note that our retriever can also be trained to obtain improved embeddings and performance; however, we specifically aimed to demonstrate an algorithm that can be deployed off-the-shelf without any training in data-scarce setups (precisely like top-k, which accounts for its widespread popularity in practice).
>
> We appreciate the reviewer pointing out these relevant papers and will ensure they are properly cited with appropriate discussion regarding methods that involve training.

---

> ### Author Response · Authors · 2025-11-22
> **Rebuttal by Authors (Part 2)**
>
> **Table 1:** Accuracy (%) on SST-2 and CMSQA when scaling from 1.5B/3B models (Qwen-2.5-1.5B, Llama3.2-3B) to 7B/8B models (Qwen-2.5-7B, Llama3-8B). KITE consistently improves over zero-shot, random, Dense, and DPP baselines across all model sizes.
>
> | Method    | SST-2 Qwen-2.5-1.5B | SST-2 Qwen-2.5-7B | SST-2 Llama3.2-3B | SST-2 Llama3-8B | CMSQA Qwen-2.5-1.5B | CMSQA Qwen-2.5-7B | CMSQA Llama3.2-3B | CMSQA Llama3-8B |
> |-----------|----------------------|---------------------|---------------------|--------------------|-----------------------|----------------------|----------------------|---------------------|
> | Zero-shot | 61.69               | 65.82              | 68.73              | 74.21             | 57.00                | 64.53               | 58.86               | 61.52              |
> | Random    | 63.36               | 71.53              | 89.84              | 90.12             | 65.90                | 68.29               | 67.22               | 68.95              |
> | Dense     | 88.99               | 89.45              | 92.55              | 93.35             | 70.18                | 72.35               | 72.23               | 74.19              |
> | DPP       | 90.25               | 91.63              | 91.86              | 93.61             | 70.05                | 74.82               | 71.65               | 73.83              |
> | KITE      | **93.35**           | **94.50**          | **94.28**          | **94.84**         | **71.25**            | **75.69**           | **72.89**           | **75.22**          |
>
>
> > The focus is primarily on classification tasks
>
> We acknowledge this limitation. Extending our theoretical framework to more complex generation tasks represents a crucial direction for future research.
>
> > The proposed approach is model-agnostic, meaning that the exemplars are selected only using BERT without knowing the target LLM.
>
> Indeed, this is correct. We respectfully submit that this constitutes a strength of our work rather than a limitation, as it enables broad applicability across different LLM architectures without requiring model-specific adaptations.
>
>
> > Computational scalability
>
> The computational cost of our algorithm with an embedding function of dimension $d$,  an example bank of $T$ examples, and $k$ exemplars is \(O($k T d^{2}$)\). The computation costs are linear in $k$ and $T$, and only quadratic in the dimension $d$ (due to the Sherman–Morrison trick). We note that the computational cost of the top-k algorithm is $O(d T)$. Hence, if $k$ is small (typically a constant in practice), the computational cost of our algorithm is only a factor of $d$ larger than that of the classic top-k retrieval algorithm.
>
> > How does the model perform if the embedding is changed from BERT to other embeddings, such as E5 or RoBERTa?
>
> We have conducted experiments to demonstrate that the relative performance of our method (compared to baselines) remains consistent across different embeddings. Please refer to Table 2, where we evaluate bert-base-uncased, RoBERTa-base, e5-base-v2, DistilBERT, and MiniLM. KITE consistently matches or outperforms Dense and DPP across all these encoders.
>
> **Table 2:** Accuracy (%) on SST-5 with Qwen-2.5-1.5B using different embedding encoders for retrieval. KITE consistently matches or improves upon Dense and DPP across all encoders.
>
> | Method | bert-base-uncased | roberta-base | e5-base-v2 | distilbert-base-uncased | all-MiniLM-L6-v2 |
> |--------|-------------------|--------------|------------|--------------------------|-------------------|
> | DPP | 47.88 | 41.41 | 45.68 | 48.77 | 48.13 |
> | Dense | 46.64 | 36.33 | 45.14 | 49.04 | 48.68 |
> | KITE | **49.59** | **48.31** | **48.04** | **49.29** | **48.72** |
>
>
> > It is known that small LLMs exhibit weaker in-context learning ability. Will the method work for larger models?
>
> We have added experiments with 7B/8B-scale models; these results are presented in Table 1. We observe that larger models naturally achieve higher absolute performance than their smaller counterparts, and KITE consistently improves over all baselines for both small and large models.
>
> > It is unclear how the approach works for multi-class classification. Does the author have one kernel for all classes, or each class has its own kernel?
>
> We appreciate the opportunity to clarify this point. We wish to note that the paper does not restrict the methodology to binary classification. In fact, most of the datasets used in our experiments involve multi-class settings where one correct answer must be selected among several options. Therefore, the results presented in the paper are for multi-class classification problems, where a single kernel is used across all classes. We apologize if this was not sufficiently clear in the original manuscript.

---

> > ### Author Response · Authors · 2025-11-25
> > **Request to Reviewer zKVx**
> >
> > Dear Reviewer zKVx,
> >
> > We sincerely thank you again for your valuable feedback, which has helped us improve the quality of our work. As the discussion deadline is approaching, we kindly request you to review our rebuttal.
> >
> > We are happy to address any remaining concerns. If our responses adequately address your concerns, we kindly ask you to consider adjusting the corresponding scores. Thank you again for your time and effort!
> >
> > Regards,
> >
> > Authors

---

> > > ### Comment · Reviewer_zKVx · 2025-11-27
> > >
> > > Appreciate the authors for the rebuttal, which addresses many of my concerns. I recognize the distinctions between KITE and [1][2], and I have increased my scores accordingly. However, I still stand by some of my stands.
> > >
> > > W1.1
> > > - The theoretical understanding does not translate into meaningful numbers or suggestions, such as how the hyperparameters should be configured in practice.
> > > - Having a submodularity ratio alone for the algorithm is insufficient to demonstrate its effectiveness, since one can also argue that top-k retrieval also has a submodularity ratio. The argument would be stronger if the submodularity ratios could be compared, or if it could be shown that perhaps top-k retrieval does not possess such a property.
> > > - If put into a non-LLM context, the theoretical analysis has insufficient reference to existing theoretical works on Sequential Decision Making, Bandit algorithms, and Bayesian Optimization, where submodularity is commonly discussed.
> > >
> > > W1.2 Although the word submodularity appeared once in [1], it uses the NeuralUCB algorithm with greedy sample selection, which is fundamentally similar to the approximate submodularity discussed in this paper.
> > >
> > > Q1. I noticed that the authors set k=50 in practice, which does not seem to be negligible. It would be helpful if the actual computational time could be documented.

---

> > > > ### Author Response · Authors · 2025-11-29
> > > > **Response to Reviewer Comments**
> > > >
> > > > We thank the reviewer for their reply and feedback. We believe there may be some confusion regarding the nature of submodularity ratios and their relationship to algorithms, which we address below:
> > > >
> > > > **On Submodularity Ratios and Algorithm Comparisons** We respectfully clarify two important points:
> > > >
> > > > 1. *Submodularity ratios are properties of set functions, not algorithms*. The submodularity ratio (Definition 2) is defined for a set function $f:2^{\mathcal{X}} \rightarrow \mathbb{R}$. We establish an approximate submodularity ratio for our objective function (Equation 4) in Lemma 1. It is not meaningful to discuss or compare the "submodularity ratios" of KITE and top-k retrieval as algorithms; these ratios characterize objective functions, not selection procedures. Since KITE and top-k optimize different objectives, theoretical comparison via submodularity ratios is not applicable, and comparison must be empirical.
> > > >
> > > > 2. *Greedy optimization does not imply submodularity properties*. The relationship is the converse: establishing a submodularity ratio for a function f provides approximation guarantees for greedy algorithms optimizing f. In [1], while the objective (Equation 1) is indeed a set function over the example bank, no submodularity guarantees are established. Moreover, we argue that characterizing the submodularity ratio of their objective is infeasible due to the opaque nature of the black-box LLM $f(\cdot)$, which the authors themselves acknowledge is unattainable. We therefore respectfully disagree that NeuralUCB is "fundamentally similar to the approximate submodularity discussed in this paper."
> > > >
> > > > **Our key contribution** is establishing the analogous problem within a kernelized linear model framework and deriving the optimization objective from first principles—this theoretical grounding is what enables our submodularity analysis.
> > > >
> > > > **On Hyperparameter Configuration.** We refer the reviewer to our extensive ablation studies (Lines 443–455 and Appendix Section A), which detail our hyperparameter selection process. While Bayesian optimization methods exist for hyperparameter tuning (e.g., Shahriari et al., 2016), this is orthogonal to our contributions.
> > > >
> > > > **On References to Related Theoretical Work.** Submodular optimization is a broad and well-established field. Sequential decision making, bandit algorithms, and Bayesian optimization, while related areas where submodularity appears, are not directly connected to our specific contributions. We have cited standard references on these topics:
> > > >
> > > > + Bandits/Sequential Decision Making: Lattimore & Szepesvári (2020), Bandit Algorithms, Cambridge University Press.
> > > >
> > > > + Optimal Design: Pukelsheim (2006), Optimal Design of Experiments, SIAM.
> > > >
> > > > + Submodular Optimization: Das & Kempe (2011), "Submodular meets spectral: Greedy algorithms for subset selection, sparse approximation, and dictionary selection."
> > > >
> > > > We are happy to include additional references if the reviewer can suggest specific works they believe are missing.
> > > >
> > > >
> > > > **Response to Q1: Computational Cost** All baselines (BM25, Dense, DPP, KITE) use $k=50$ in-context examples per query, resulting in similar prompt lengths. Consequently, LLM inference dominates the computational cost, and the choice of selection strategy has minimal impact on end-to-end latency.
> > > >
> > > > Wall-clock time per query on SST-5 (8,534 example bank, 1,101 validation queries):
> > > > | Method | Qwen-2.5-7B (s) | Qwen-2.5-1.5B (s) |
> > > > |--------|-----------------|-------------------|
> > > > | BM25   | 2.42            | 0.20              |
> > > > | Dense  | 2.45            | 0.21              |
> > > > | DPP    | 2.58            | 0.21              |
> > > > | KITE   | 2.61            | 0.23              |
> > > >
> > > > We observe that KITE introduces minimal overhead compared to standard dense retrieval, leaving overall pipeline latency effectively unchanged.
> > > >
> > > >
> > > >
> > > > [1] Wu, Z., Lin, X., Dai, Z., Hu, W., Shu, Y., Ng, S. K., ... & Low, B. K. H. (2024). Prompt optimization with EASE? efficient ordering-aware automated selection of exemplars. Advances in Neural Information Processing Systems, 37, 122706-122740.

---

### Comment · Area_Chair_mqJQ · 2025-11-27
**Reviewer Reminder: Author Rebuttals Available**

Dear Reviewers,

The authors have posted their rebuttals to your reviews.

Please read the authors' responses, assess whether your concerns have been addressed, and update your ratings accordingly.

Your prompt attention to the rebuttals is appreciated.

Best,
AC

---

### Note · Program_Chairs · 2026-01-17
**Submission Desk Rejected by Program Chairs**

The following references in this submission do not refer to real documents and/or have major errors in bibliographic information:

 Madeleine Sorensen, Nan Ding, and Ming-Wei Chang. Information-theoretic demonstration selection for few-shot prompting. In Proceedings of the Findings of EMNLP, 2022. 3"